# Latent Generative Models with Tunable Complexity for Compressed Sensing and other Inverse Problems

## Abstract

Generative models have emerged as powerful priors for solving inverse problems. These models typically represent a class of natural signals using a single fixed complexity or dimensionality. This can be limiting: depending on the problem, a fixed complexity may result in high representation error if too small, or overfitting to noise if too large. We develop tunable-complexity priors for diffusion models, normalizing flows, and variational autoencoders, leveraging nested dropout. Across tasks including compressed sensing, inpainting, denoising, and phase retrieval, we show empirically that tunable priors consistently achieve lower reconstruction errors than fixed-complexity baselines. In the linear denoising setting, we provide a theoretical analysis that explicitly characterizes how the optimal tuning parameter depends on noise and model structure. This work demonstrates the potential of tunable-complexity generative priors and motivates both the development of supporting theory and their application across a wide range of inverse problems.

## 1 Introduction

Inverse problems aim to reconstruct an unknown signal, potentially corrupted by noise, from a set of measurements given by a forward model, which may or may not be known in advance. Such a formulation applies to various image-processing applications, including compressive sensing, denoising, and super-resolution. In practice, inverse problems are ill-posed and thus they require prior information about the signal to yield a successful recovery (Scarlett et al., 2022).

Deep generative models have been demonstrated to be powerful signal priors when used to solve inverse problems (Bora et al., 2017; Hand et al., 2018; Asim et al., 2020; Daras et al., 2021; 2022; Song et al., 2022). The process of using a deep generative prior typically has two phases: training and inversion. In the training phase, a generative neural network is trained on a dataset representative of the natural signal class intended for inversion. In the second phase, the model parameters are fixed from training, and an algorithm is deployed to estimate the signal of interest for a given forward operator. This approach has the benefit that the prior can be learned in isolation from the inverse problems being solved. Thus, the approach can apply to a variety of inverse problems, which is in contrast to other neural network-based approaches such as end-to-end training.

In recent years, there has been a dominant framework for using generative models as priors for inverse problems. Within this framework, generative priors have a fixed complexity that is set during training. For example, existing priors include Generative Adversarial Networks (GANs), which typically have a latent space of fixed low dimensionality (Bora et al. (2017); Hand et al. (2018); Menon et al. (2020); Cocola et al. (2020); Hand & Joshi (2019)); Normalizing Flow models, which have a latent space of fixed high-dimensionality equal to that of the images (Asim et al. (2020); Whang et al. (2021); Ardizzone et al. (2019); Liu et al. (2023)); and Score-Based Models, which maintain information about the probability density over the fixed high-dimensional space of all images (Song et al. (2022); Meng & Kabashima (2023); Jalal et al. (2021); Dou & Song (2024); Chung et al. (2023)).

We step outside the dominant framework and demonstrate the benefits of solving inverse problems using generative priors whose complexity can be selected by the user at inference time, after training. Such generative priors simultaneously maintain representations of varying complexities of the natural

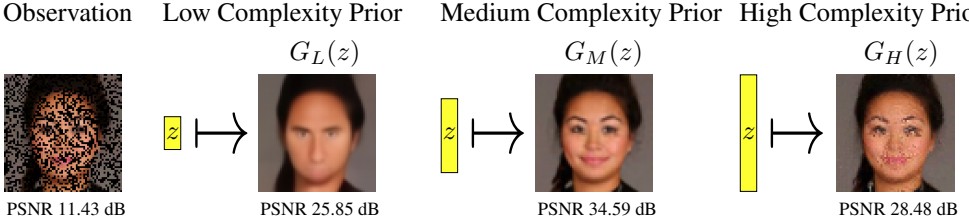

Figure 1: For visualization purposes, consider an inpainting problem with randomly missing pixels. We trained three separate generative models: one with low, medium, and high latent dimensionality as depicted by the size of the boxes representing $z$. The medium-complexity prior yields the reconstruction with the highest Peak Signal-to-Noise Ratio (PSNR).

signal class, and we refer to them as **generative priors with a tunable complexity**. *In this paper, we demonstrate that tunable generative priors can lead to significantly improved reconstruction errors when model complexity is appropriately tuned for a specific inverse problem. We focus on latent generative models whose complexity is governed by the latent dimension $k$. By training the model to preserve meaningful representations across different values of $k$, we enable practitioners to tune its complexity to the inverse problem at hand.* We illustrate this point in Figure 1 for a random pixel inpainting problem. The medium-complexity prior yields a higher quality image reconstruction than the low- and high-complexity prior, both qualitatively and quantitatively.

To motivate our approach, we first consider injective flows (Ross & Cresswell, 2021). We train a family of models, each with a distinct latent dimensionality $k$ and no parameter sharing, and apply them to compressed sensing with random measurements. As shown in Figure 2, reconstruction error follows an upside-down U-shaped curve: with fewer measurements, models of intermediate complexity outperform both higher- and lower-complexity ones. Moreover, the optimal complexity depends on the number of available measurements. While this experiment requires training separate models for each $k$, a naïve and computationally expensive strategy, it illustrates the importance of tunability. This motivates the more efficient algorithms we develop next, which scale to practical settings such as production-size images.

Building on this motivation, we will show how tunability can be achieved efficiently in three major classes of generative models: variational autoencoders (VAEs) (Kingma & Welling, 2014), normalizing flows (NFs) (Durkan et al., 2019), and latent diffusion models (LDMs) (Rombach et al., 2022). In each case, we design a single family of models with parameter sharing and provide an efficient training procedure. For LDMs, we introduce a new algorithm based on nested dropout (Rippel et al., 2014), described in Section 3.1. For NFs, we adopt an existing ordering method (Bekasov & Murray, 2020) Appendix B. For VAEs, we extend the adversarial objective of Esser et al. (2021) by adding a regularization term. Across all three settings, we empirically demonstrate that tunable generative priors consistently achieve lower reconstruction errors than fixed-complexity baselines over a range of inverse problems, undersampling ratios, and noise levels (Figure 4).

As an initial theoretical exploration into the effect of model tunability, we study denoising in the case of an invertible linear generative model and the best lower-dimensional linear models that approximate it. We rigorously establish a theory for how to select the tuning parameter in the simple case of a linear generative model. In this setting, we provide an explicit expression for the reconstruction error as a function of the modeled dimensionality in the cases of maximum likelihood estimation and maximum *a posteriori* estimation. This analytical expression permits a direct expression for optimal signal complexity in this setting, revealing the theoretical benefit for tunability.

The results in this paper demonstrate the benefits of using generative priors with a tunable complexity for inverse problems both empirically and theoretically. We empirically demonstrate the affects of tunable complexity on using families of generative models and different model architectures and inversion algorithms . Additionally, for linear models, we provide a theoretical analysis of denoising. This work motivates research on how to bring tunability into generative models.

**Our main contributions are as follows:**

- We observe a potentially surprising phenomenon in the use of latent generative priors for solving inverse problems. We train a single generative model to simultaneously represent the natural signal class across multiple latent dimensionalities $k$. Across a variety of architectures, inverse problems, and inversion algorithms, we empirically find that some intermediate latent dimensionality yields an improved reconstruction error.

- We propose a new training algorithm for latent diffusion that utilizes nested dropout and a convex combination of the original and truncated latent objective. This yields a tunable latent diffusion model that learns hierarchical representations across latent dimensionalities, enabling a single model to be tuned as a prior for downstream inverse problems.

- We provide a theoretical analysis of tunability in the context of denoising with linear invertible generative models. Under this model, we derive an explicit expression for the reconstruction error of MLE and MAP estimators as a function of the model complexity.

## 2 Background and Related Work

The background will focus on diffusion models due to their relevance in recent research trends. For additional details on normalizing flows and variational autoencoders, we refer the reader to the appropriate references.

### 2.1 Diffusion Models

The standard diffusion framework gradually corrupts data $\boldsymbol{x}_0 \sim p_{\text{data}}$, $\boldsymbol{x} \in \mathbb{R}^n$ into noise $\boldsymbol{x}_T \sim \mathcal{N}(0, \boldsymbol{I})$ via a forward noising process, and learns a parameterization of the reverse process to recover the data distribution (Ho et al., 2020; Song et al., 2021b; Dhariwal & Nichol, 2021). We adopt latent diffusion models (LDMs) (Rombach et al., 2022), which improve efficiency by running diffusion in a learned latent space: a *variational autoencoder (VAE)* (Kingma & Welling, 2014) with encoder $\mathcal{E} : \mathbb{R}^n \to \mathbb{R}^k$ and decoder $\mathcal{D} : \mathbb{R}^k \to \mathbb{R}^n$ (typically $k \ll n$) yields $\boldsymbol{z}_0 = \mathcal{E}(\boldsymbol{x}_0)$ and $\boldsymbol{x}_0 \approx \mathcal{D}(\boldsymbol{z}_0)$ for all $\boldsymbol{x}_0 \sim p_{\text{data}}$. The encoder pushes $p_{\text{data}}$ to $p_Z := \mathcal{E}_{\#} p_{\text{data}}$, and diffusion operates on $\boldsymbol{z}_t$.

With schedule $\{\beta_t\}_{t=1}^T$, $\alpha_t := 1 - \beta_t$, and $\bar{\alpha}_t := \prod_{j=1}^t \alpha_j$, the forward marginal in latent space is

$$\boldsymbol{z}_t = \sqrt{\bar{\alpha}_t}\, \boldsymbol{z}_0 + \sqrt{1 - \bar{\alpha}_t}\, \boldsymbol{\epsilon}, \qquad \boldsymbol{\epsilon} \sim \mathcal{N}(0, \boldsymbol{I}), \qquad \text{(forward marginal)}$$

which allows sampling $\boldsymbol{z}_t$ at any timestep $t$ directly from $\boldsymbol{z}_0$. The reverse process is parameterized by a neural network:

$$\boldsymbol{z}_{t-1} = \frac{1}{\sqrt{\alpha_t}}\Big(\boldsymbol{z}_t - \frac{1-\alpha_t}{\sqrt{1-\bar{\alpha}_t}}\, \boldsymbol{\epsilon}_\theta(\boldsymbol{z}_t, t)\Big) + \sigma_t\, \boldsymbol{\epsilon}_t', \qquad \boldsymbol{\epsilon}_t' \sim \mathcal{N}(0, \boldsymbol{I}), \qquad \text{(reverse)}$$

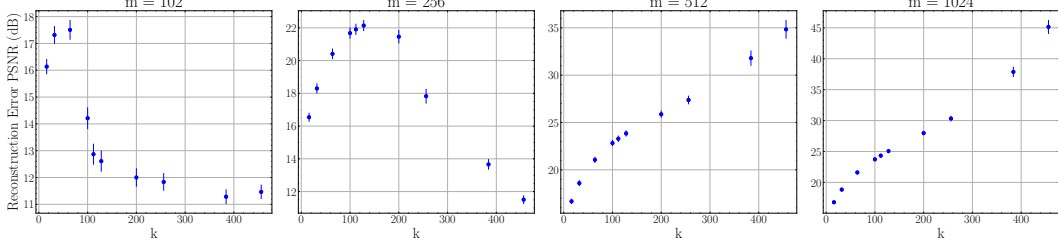

Figure 2: In this brute-force approach, we employ an injective flow prior trained separately on MNIST, where each image has size $n = 32 \times 32 = 1024$ pixels and the latent dimensionality $k$ varies from 16 to 456. For compressed sensing with random measurements, intermediate latent dimensions ($150 \le k \le 300$) yield the lowest reconstruction error at small measurement ratios $m/n$.

where $\sigma_t \in [0, \ \sigma_t^{\text{DDPM}}]$ controls sampling stochasticity and $\sigma_t^{\text{DDPM}} = \sqrt{\frac{1-\bar{\alpha}_{t-1}}{1-\bar{\alpha}_t}\left(1-\alpha_t\right)}$. Setting $\sigma_t = 0$ yields the deterministic denoising diffusion implicit model (DDIM) sampler (Song et al., 2021a).

A denoising network $\epsilon_\theta$ (typically a U-Net) is trained to predict the noise in the latent space:

$$\mathcal{L}_{\text{LDM}} = \mathbb{E}_{t \sim \mathcal{U}(\{1,\dots,T\}), \, \boldsymbol{z}_0, \, \boldsymbol{\epsilon} \sim \mathcal{N}(0,\boldsymbol{I})} \big\| \boldsymbol{\epsilon} - \epsilon_\theta\big(\sqrt{\bar{\alpha}_t}\,\boldsymbol{z}_0 + \sqrt{1-\bar{\alpha}_t}\,\boldsymbol{\epsilon}, \, t\big)\big\|_2^2, \tag{1}$$

where $t \sim \mathcal{U}(\{1,\dots,T\})$ denotes that the timestep is sampled uniformly from all diffusion steps.

To enable latent diffusion, the autoencoder must be pretrained; its objective combines reconstruction, regularization, and adversarial terms. The primary term is the reconstruction loss, $\mathcal{L}_{\text{recon}}$, typically a combination of $\ell_1$ and perceptual losses, which measures the discrepancy between the original high-dimensional signal $\boldsymbol{x}$ and its reconstruction $\mathcal{D}(\mathcal{E}(\boldsymbol{x}))$. A regularization penalty, $\mathcal{L}_{\text{reg}}$, is enforced via Kullback–Leibler (KL) divergence with respect to a reference distribution, typically the standard Gaussian. Finally, an adversarial loss, $\mathcal{L}_{\text{adv}}$, is introduced by training a discriminator $\mathcal{C}$ to distinguish between real images $\boldsymbol{x}$ and reconstructed samples $\mathcal{D}(\mathcal{E}(\boldsymbol{x}))$

$$\mathcal{L}_{\text{Autoencoder}} = \mathcal{L}_{\text{rec}}\big(x, \, \mathcal{D}(\mathcal{E}(x))\big) + \lambda_{\text{reg}}\,\mathcal{L}_{\text{reg}}\big(\mathcal{E}(x)\big) + \lambda_{\text{adv}}\,\mathcal{L}_{\text{adv}}\big(\mathcal{C}, \, x, \, \mathcal{E}(x)\big). \tag{2}$$

After this stage, diffusion is performed entirely on $\boldsymbol{z}_t$.

## 2.2 Inverse Problems with Generative Priors

We consider the general linear inverse problem of recovering an unknown signal $\boldsymbol{x} \in \mathbb{R}^n$ from noisy measurements $\boldsymbol{y} \in \mathbb{R}^m$,

$$\boldsymbol{y} = \mathcal{A}(\boldsymbol{x}) + \boldsymbol{\eta}, \quad \boldsymbol{\eta} \sim \mathcal{N}(0, \boldsymbol{\sigma^2 I}_m), \tag{3}$$

where $\mathcal{A} : \mathbb{R}^n \to \mathbb{R}^m$ is the forward operator (e.g., a linear projection, convolutional blur, or other transformation) (Scarlett et al., 2022), and $\boldsymbol{\eta}$ is additive noise drawn from a Gaussian distribution. Generative models can be employed as priors for inverse problems in two main ways: supervised, where the forward operator is known during training and models are trained on paired data $\{(x_i, y_i)\}$, or unsupervised, where the forward operator is unknown (Scarlett et al., 2022; Li et al., 2025).

For inverse problems, we modify the reverse dynamics in Equation (reverse) to account for measurements $\boldsymbol{y} = \mathcal{A}(\boldsymbol{x}) + \boldsymbol{\eta}$. Since we operate entirely in latent space, posterior sampling is guided by the conditional score, which decomposes as

$$\nabla_{\boldsymbol{z}_t} \log p_t(\boldsymbol{z}_t \mid \boldsymbol{y}) = \nabla_{\boldsymbol{z}_t} \log p_t(\boldsymbol{z}_t) + \nabla_{\boldsymbol{z}_t} \log p_t(\boldsymbol{y} \mid \boldsymbol{z}_t). \tag{4}$$

The first term is provided by the pretrained diffusion prior through the denoising network $\epsilon_\theta$, while the second term enforces data-consistency with the forward operator $\mathcal{A}$ under the measurement model.

Following the taxonomy of interleaving methods highlighted by Wang et al. (2024), these solvers alternate unconditional reverse diffusion steps (e.g., DDIM) with data-consistency corrections that either explicitly approximate the measurement likelihood or enforce feasibility. One line of methods explicitly approximates the measurement-likelihood term via a projection or gradient update (Jalal et al., 2021; Kawar et al., 2021; Chung et al., 2023; Wang et al., 2023; Rout et al., 2024). A canonical formulation is latent diffusion posterior sampling (Rout et al., 2023):

$$\nabla_{\boldsymbol{z}_t} \log p(\boldsymbol{y} \mid \boldsymbol{z}_t) \ \approx \ \nabla_{\boldsymbol{z}_t} \log p(\boldsymbol{y} \mid \boldsymbol{x}_0 = \mathcal{D}(\mathbb{E}\left[z_0 \mid z_t\right])) = \tfrac{1}{\sigma^2}\,\nabla_{\boldsymbol{z}_t}\,\|\boldsymbol{y} - \mathcal{D}(\boldsymbol{z}_0)\|_2^2, \tag{5}$$

where the posterior mean is given by Tweedie's formula (Efron, 2011).

Another class of methods does not explicitly compute the measurement likelihood, but instead approximates the feasible set of solutions $\{\boldsymbol{x} \mid \boldsymbol{y} = \mathcal{A}(\boldsymbol{x})\}$ (Kawar et al., 2021; 2022; Song et al., 2024). In particular Song et al. (2024) employs a hard data-consistency term that enforces $\mathcal{D}(\boldsymbol{z}) \in \{\boldsymbol{x} \mid \boldsymbol{y} = \mathcal{A}(\boldsymbol{x})\}$, which is approximated in practice by gradient-descent.

# 3 Methods

This section describes the training of a tunable latent diffusion model (Section 3.1) and its application as a prior for inverse problems (Section 3.1). We begin by training a variational autoencoder (VAE)

as the backbone of the latent diffusion model. The VAE consists of an encoder $\mathcal{E} : \mathbb{R}^n \to \mathbb{R}^k$ and a decoder $\mathcal{D} : \mathbb{R}^k \to \mathbb{R}^n$, such that for samples $\boldsymbol{x}_0 \sim p_{\text{data}}(\boldsymbol{x}_0)$ we have $\boldsymbol{x}_0 \approx \mathcal{D}(\mathcal{E}(\boldsymbol{x}_0))$. Once trained the autoencoder, a denoising network $\boldsymbol{\epsilon}_\theta(\boldsymbol{z}_t, t)$ is trained in latent space to predict the Gaussian noise at each diffusion step $t$. Finally, diffusion is performed in latent space using the denoising network $\boldsymbol{\epsilon}_\theta$.

## 3.1 Training a Tunable Latent Diffusion

We aim to train a latent generative model that can represent the natural signal class across multiple latent dimensionalities. To achieve this, we leverage nested dropout (Rippel et al., 2014), which imposes an ordered structure on the latent variables by always preserving a prefix of coordinates. Formally, let $k \sim p_k$ be drawn from $\{1, \ldots, d\}$. In our experiments, we use a truncated geometric distribution with success parameter $p$, but other distribution over $\{1, \ldots, d\}$ can be adopted depending on the application. The truncation operator is defined as $\boldsymbol{z}_{\downarrow k} = [\boldsymbol{z}_1, \boldsymbol{z}_2, \ldots, \boldsymbol{z}_k, 0, \ldots, 0]$ with $k \le d$. Given $\boldsymbol{z} = \mathcal{E}(\boldsymbol{x})$, reconstruction is performed from $\boldsymbol{z}_{\downarrow k}$, which encourages earlier coordinates to carry more information about the signal class.

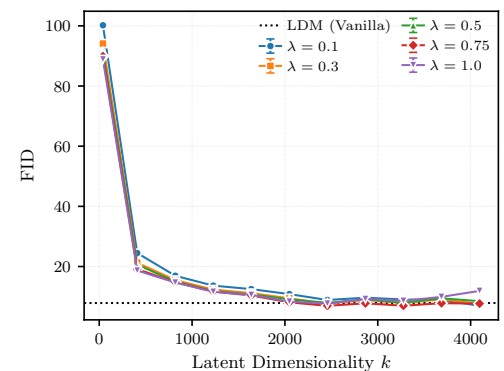

Figure 3: FID as a function of latent dimensionality, evaluated on 50k training images from CelebA. FID score computed with (Parmar et al., 2022).

Latent diffusion is trained in two stages. First, the autoencoder is trained to reconstruct $\boldsymbol{x} \in \mathbb{R}^n$ robustly across a range of latent dimensions $k$. Building on Section 2.1, the VAE backbone is trained with reconstruction, regularization, and adversarial terms, which we extend here with a nested dropout objective. This follows the standard VAE training objective with an added nested dropout term:

$$\mathcal{L}_{\text{VAE}} = \min_{E, D} \max_{\mathcal{C}} \Big[ \mathcal{L}_{\text{rec}}\big(\boldsymbol{x}, \mathcal{D}(\mathcal{E}(\boldsymbol{x}))\big) + \lambda_{\text{reg}} \, \mathcal{L}_{\text{reg}}\big(\mathcal{E}(\boldsymbol{x})\big)$$

$$+ \lambda_{\text{adv}} \, \mathcal{L}_{\text{adv}}\big(\mathcal{C}, \boldsymbol{x}, \mathcal{E}(\boldsymbol{x})\big) + \lambda_{\text{drop}} \, \mathcal{L}_{\text{drop}}\big(\boldsymbol{x}, \mathcal{D}(\mathcal{E}(\boldsymbol{x})_{\downarrow k})\big) \Big], \quad (6)$$

where $\mathcal{C} : \mathbb{R}^n \to (0, 1)$ is a discriminator. We adopt a perceptual loss (Zhang et al., 2018) for $\mathcal{L}_{\text{drop}}$.

In the second stage, the diffusion model is trained in latent space with a loss that interpolates between the standard diffusion objective and its truncated-latent variant. For $\boldsymbol{\lambda} \in [0, 1]$, we define

$$\mathcal{L}_{\text{LDM}} = \mathbb{E}_{\mathcal{E}(\boldsymbol{x}), \boldsymbol{\epsilon} \sim \mathcal{N}(0,1), t \sim \mathcal{U}(\{1, \ldots, T\})} \Big[ (1 - \boldsymbol{\lambda}) \, \|\boldsymbol{\epsilon} - \boldsymbol{\epsilon}_\theta(\boldsymbol{z}_t, t)\|_2^2 + \boldsymbol{\lambda} \, \|\boldsymbol{\epsilon} - \boldsymbol{\epsilon}_\theta((\boldsymbol{z}_t)_{\downarrow k}, t)\|_2^2 \Big]. \quad (7)$$

The first term is the standard latent diffusion objective, while the second applies it to the truncated latent $(\boldsymbol{z}_t)_{\downarrow k}$, encouraging effective denoising even from a reduced representation. Here, $\boldsymbol{z}_t$ is sampled from the forward DDPM process (Equation (forward marginal)), and $k$ is drawn from the same distribution $p_k$ used in the VAE objective, ensuring consistency. This encourages the latent space to organize hierarchically: early coordinates contain coarse information preserved under all truncation levels, while later coordinates refine the representation when higher dimensions are available. This effect is illustrated in Figure 3, which shows how FID varies with latent dimensionality under different $\boldsymbol{\lambda}$ values.

For details about training please go to Appendix A.2. For details on tunable normalizing flows and their use as priors for inverse problems, see Appendix B.

## 3.2 Inverse Problems with Tunable Priors

A broad family of methods applies a data-consistency step via projection, gradient, or small optimization after each reverse update to move the prior iterate toward the feasible set $\{x \mid \mathcal{A}(x) = y\}$. In

**Algorithm 1** General Template for Tunable Diffusion Priors

1: **Input:** $y$, $\mathcal{A}$, $\mathcal{E}$, $\mathcal{D}$, $\boldsymbol{\epsilon}_\theta$, steps $T$, tunable parameter $k$
2: **Output:** $\mathcal{D}(\hat{z}_0)$
3: $z_T \sim \mathcal{N}(0, I)$
4: **for** $t = T - 1$ **to** $0$ **do**
5:      $\hat{s} \leftarrow \boldsymbol{\epsilon}_\theta(z_t, t)$
6:      $\hat{z}_0 \leftarrow \frac{1}{\sqrt{\bar{\alpha}_t}}(z_t + \sqrt{1 - \bar{\alpha}_t}\hat{s})$
7:      $z'_{t-1} \leftarrow$ DDIM/DDPM reverse with
8:          $\hat{z}_0, \hat{s}$
9:      $z_{t-1} \leftarrow$ project/gradient update with
10:          $\hat{z}_0$ and $z'_{t-1}$ to get closer to
11:          $\{z \mid \mathcal{A}(\mathcal{D}(z)) = y\}$
12:      $z_{t-1} \leftarrow (z_{t-1})_{\downarrow k}$
13: **end for**
14: **return** $\mathcal{D}(\hat{z}_0)$

**Algorithm 2** Tunable Posterior Sampling (Concrete Instantiation)

1: **Input:** $y$, $\mathcal{A}$, $\mathcal{E}$, $\mathcal{D}$, $\boldsymbol{\epsilon}_\theta$, steps $T$, variances $\{\sigma_t\}$, tunable parameter $k$
2: **Output:** $\mathcal{D}(\hat{z}_0)$
3: $z_T \sim \mathcal{N}(0, I)$
4: **for** $t = T - 1$ **to** $0$ **do**
5:      $\hat{s} \leftarrow \boldsymbol{\epsilon}_\theta(z_t, t)$
6:      $\hat{z}_0 \leftarrow \frac{1}{\sqrt{\bar{\alpha}_t}}(z_t + \sqrt{1 - \bar{\alpha}_t}\hat{s})$
7:      $\boldsymbol{\epsilon} \sim \mathcal{N}(0, I)$
8:
9:      $z'_{t-1} \leftarrow \frac{\sqrt{\alpha_t(1-\bar{\alpha}_{t-1})}}{1-\bar{\alpha}_t}z_t + \frac{\sqrt{\bar{\alpha}_{t-1}}\beta_t}{1-\bar{\alpha}_t}\hat{z}_0$
10:          $+ \sigma_t \boldsymbol{\epsilon}$
11:      Initialize $z$ at $\hat{z}_0$
12:      $z_{t-1} \leftarrow \arg\min_z \| y - \mathcal{A}(\mathcal{D}(z)) \|_2^2$
13:          $+ \frac{1}{2\sigma_t^2}\| z - z'_{t-1} \|_2^2$
14:      $z_{t-1} \leftarrow (z_{t-1})_{\downarrow k}$
15: **end for**
16: **return** $\mathcal{D}(\hat{z}_0)$

latent space, representative examples include Latent Posterior Diffusion (LDPS) (Rout et al., 2023), Posterior Sampling with Latent Diffusion (PSLD) (Rout et al., 2023), ReSample (Song et al., 2024), and our formulation in Algorithm 2. The goal of Algorithm 2 is not to establish state-of-the-art performance, but rather to provide a simple and broadly effective algorithm across a variety of inverse problems. By contrast, contemporary inversion algorithms have certain drawbacks: PSLD relies on a correction term whose form is unclear for arbitrary forward operators, while ReSample enforces data consistency through an optimization that bridges latent and pixel spaces, making it unclear how to define a tunable parameter in pixel space without dedicated training. The experiments in Figure 4 employ Algorithm 2, whereas tunable versions of LDPS and PSLD would fall under the general template of Algorithm 1.

Our approach follows a generic template for tunable diffusion priors. Starting from Gaussian noise in the latent space, the algorithm iteratively denoises the latent variable using the learned noise prediction network. Each reverse step is then corrected by a data-consistency operation that is consistent with measurements $y = \mathcal{A}(\mathcal{D}(z))$. To tune the latent dimensionality across different dimensionality truncation operator at every iteration. This template unifies a broad family of inversion methods:

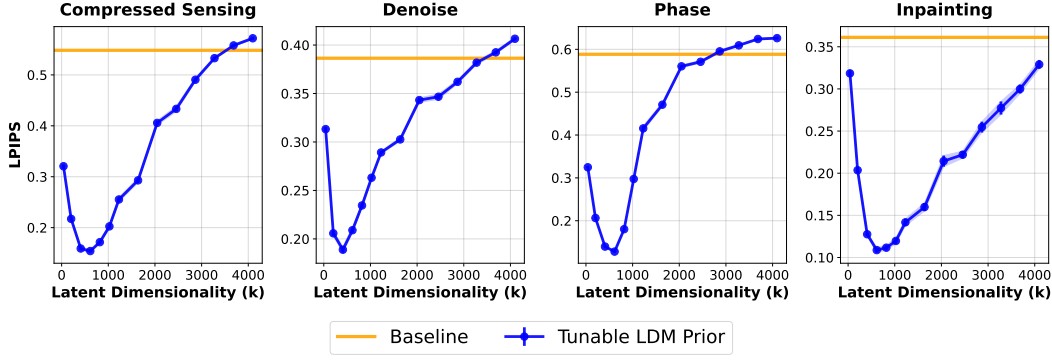

Figure 4: Performance of generative prior with a tunable complexity (Tunable LDM Prior) and its fixed complexity counterpart (BASELINE) for compressed sensing, inpainting, and denoising on CelebA Dataset. Tunable prior demonstrates a range of tunable parameters $k$ that lead to a better estimate of the target signal measured in LPIPS than its baseline.

the prior supplies a generative update, the data-consistency step enforces measurements, and the tunable operator controls the representation capacity. The concrete instantiation in Algorithm 2 implements this template by combining a DDPM/DDIM reverse update with a quadratic data-consistency optimization.

# 4 Experiments

Our experiments underscore several key findings. First, we successfully demonstrate the construction of a tunable model (Figure 8). Second, tunable priors are broadly applicable they are not limited to any specific architecture, class of generative models, or inversion algorithm (Figures 2 and 4, Appendix B, and Tables 1 and 2). Finally, our approach is competitive with contemporary strong baselines (Dou & Song, 2024) and further improves upon already successful priors (Rout et al., 2023) demonstrated in Tables 1 and 2. The code will be released upon after the review period.

**Naive Approach to Solving Tunable Prior** We trained a family of injective flows following Ross *et al.* Ross & Cresswell (2021) with latent dimensionalities 16, 32, 64, 100, 112, 128, 200, 256, 384, 456, each on MNIST ($N = 1024$) for 1,000 epochs using their official GitHub Repository, without parameter sharing. All models had the same number of parameters and were used as priors for compressed sensing with Gaussian random measurements $A \in \mathbb{R}^{m \times n}$ and additive Gaussian noise $\eta$ normalized to $\sqrt{\mathbb{E}|\eta|^2} = 0.1$.

**Training Details** To ensure reproducibility, we built on the Diffusers package (von Platen et al., 2022). For the VAE, we trained a standard continuous variant, while for the LDM we employed a U-Net in half-precision (float16) for memory efficiency, with approximately 200 million parameters operating on a $16 \times 16 \times 16$ latent space for CelebA (Coates et al., 2011) at an image size of $64 \times 64 \times 3$. All models were trained on a single RTX 2080 Ti GPU; training required about seven days in total—two days for the VAE and five days for the LDM. We used the AdamW optimizer (Loshchilov & Hutter, 2019) with a learning rate of $1 \times 10^{-4}$ and a batch size of 128. Images were resized to $64 \times 64$ and normalized to $[-1, 1]$. Lastly, based on our objective, we found that setting $\alpha = 0.75$ yielded the lowest FID when averaged across latent dimensionalities. We therefore selected this model for solving inverse problems.

**Inverse Problem Details** For Figure 4, we evaluated four forward operators. All experiments were done on 64x64x3 Compressed sensing used a random Gaussian operator $A \in \mathbb{R}^{m \times n}$ with i.i.d. $\mathcal{N}(0, 1/m)$ entries, where $m = 1228$ (approximately 10% of $n = 12288$). Denoising was modeled as additive Gaussian noise $\boldsymbol{\eta} \sim \mathcal{N}(0, \sigma^2 I_n)$ with $\sigma = 0.25$. Inpainting employed a binary mask $M$ with approximately 80% missing pixels. Phase retrieval used phaseless Gaussian measurements $|Ax|$, sharing the same $A$ as compressed sensing, with a measurement ratio $m/n = 0.15$ (15%). All inverse problems were solved using Algorithm 2. In Table 1, we compare our approach to a contemporary state-of-the-art (SOTA) pixel-based diffusion prior (Chung et al., 2023) and show that our framework can also enhance other SOTA priors, such as PSLD (Rout et al., 2023), by leveraging

| Method | CS | | PHASE | |
|---|---|---|---|---|
| | PSNR ↑ | LPIPS ↓ | PSNR ↑ | LPIPS ↓ |
| DPS (Chung et al., 2023) | $25.65 \pm 2.67$ | $\mathbf{0.1593 \pm 0.0670}$ | $22.15 \pm 3.84$ | $0.2665 \pm 0.0612$ |
| Tunable LDM Prior (ours) | $25.49 \pm 1.70$ | $0.1790 \pm 0.0451$ | $25.24 \pm 1.71$ | $\mathbf{0.1681 \pm 0.0464}$ |
| LDPS (Rout et al., 2023) | $23.30 \pm 1.24$ | $0.2478 \pm 0.0634$ | $24.48 \pm 1.35$ | $0.1691 \pm 0.0381$ |
| Tunable LDPS (ours) | $24.69 \pm 1.37$ | $0.1635 \pm 0.0360$ | $21.95 \pm 1.31$ | $0.2564 \pm 0.0652$ |
| NF (Asim et al., 2020) | $21.31 \pm 1.94$ | $0.5080 \pm 0.0793$ | $20.67 \pm 2.32$ | $0.5544 \pm 0.0729$ |
| Tunable NF (ours) | $\mathbf{27.16 \pm 1.56}$ | $0.2375 \pm 0.0582$ | $\mathbf{26.11 \pm 2.07}$ | $0.2457 \pm 0.0758$ |

Table 1: Quantitative Results on CelebA-HQ for Compressive Sensing and Phase Retrieval.

tunability to improve reconstruction quality. In addition, we introduce a tunable generative prior based on normalizing flows; details are provided in Appendix B. This model is trained following the objective of (Bekasov & Murray, 2020) and adopts the MAP estimator of Asim et al. (2020). Methods designated as tunable fall under our general formulation (Algorithm 1), and our goal is to enable a broad class of latent diffusion–based inversion algorithms to achieve improved results. For the measurement operators, we consider: compressed sensing with a measurement ratio $m/n = 0.075$; phase retrieval with $10\%$ measurements; inpainting with $80\%$ of pixels missing at random; and Gaussian deblurring with kernel size 5 and standard deviation 3. Lastly, for DPS we use a SOTA diffusion model trained on CelebA that achieves an FID of 1.27 on the training set (Ning et al., 2023).

| | Inpaint (random) | | Deblur (Gauss) | |
|---|---|---|---|---|
| **Method** | PSNR ↑ | LPIPS ↓ | PSNR ↑ | LPIPS ↓ |
| DPS (Chung et al., 2023) | $23.18 \pm 2.11$ | $0.1456 \pm 0.0420$ | $27.90 \pm 1.53$ | $0.0927 \pm 0.0187$ |
| Tunable LDM Prior (Ours) | $25.30 \pm 1.76$ | $0.1221 \pm 0.0280$ | $\mathbf{29.52 \pm 1.11}$ | $\mathbf{0.0852 \pm 0.0239}$ |
| PSLD (Rout et al., 2023) | $25.85 \pm 2.18$ | $0.1361 \pm 0.0360$ | $27.33 \pm 1.23$ | $0.1650 \pm 0.0371$ |
| Tunable PSLD (Ours) | $\mathbf{26.75 \pm 1.71}$ | $\mathbf{0.0946 \pm 0.0280}$ | $28.50 \pm 1.05$ | $0.1149 \pm 0.0305$ |
| NF (Asim et al., 2020) | $20.07 \pm 2.42$ | $0.3760 \pm 0.0912$ | $21.43 \pm 2.11$ | $0.4380 \pm 0.0341$ |
| Tunable NF (Ours) | $23.24 \pm 2.43$ | $0.1670 \pm 0.0569$ | $24.85 \pm 2.21$ | $0.2153 \pm 0.0433$ |

Table 2: Quantitative Results on CelebA-HQ for Inpainting and Deblur.

## 5 Theory for Denoising with Linear Generative Model

The goal of this section is to provide preliminary theoretical conclusions that show tunability can lead to improved reconstruction errors relative to corresponding nontunable models. Further, this section aims to provide justification for how to select the tuning parameter for a generative model with tunable complexity. We present these insights for the problem of denoising under additive Gaussian noise. We show that the effect of tunability can be proven even in the case of linear generative models, and thus we restrict our attention to those.

We consider a family of linear generative models given as follows. Fix an invertible $\boldsymbol{G} \in \mathbb{R}^{n \times n}$, and let $\boldsymbol{G} = \boldsymbol{U}\boldsymbol{\Sigma}\boldsymbol{V}^T$ be a singular value decomposition with $\boldsymbol{\Sigma} = \mathrm{diag}(\boldsymbol{s}_1, \dots, \boldsymbol{s}_n) \in \mathbb{R}^{n \times n}$. For any $k \leq n$, let $\boldsymbol{G}_k = \boldsymbol{U}\Sigma_k \boldsymbol{V}^T \in \mathbb{R}^{n \times n}$, where $\Sigma_k = \mathrm{diag}(\boldsymbol{s}_1, \dots, \boldsymbol{s}_k, 0, \dots, 0) \in \mathbb{R}^{n \times n}$. Note that $\boldsymbol{G} = \boldsymbol{G}_n$. Each $\boldsymbol{G}_k$ induces a probability distribution over $\mathbb{R}^n$ by

$$\boldsymbol{x} = \boldsymbol{G}_k \boldsymbol{z}, \text{ where } \boldsymbol{z} \sim \mathcal{N}(0, \boldsymbol{I}_n).$$

We will refer to this distribution as $p_{\boldsymbol{G}_k}$ and observe that $p_{\boldsymbol{G}_k} = \mathcal{N}(0, \boldsymbol{G}_k \boldsymbol{G}_k^T)$. Note that $k$ is a parameter that governs the complexity of the modeled signal class.

We consider the following denoising problem. Let $y = x_0 + \eta$, where $\boldsymbol{x}_0 \sim p_{\boldsymbol{G}_n}$ and $\eta \sim \mathcal{N}(\boldsymbol{0}, \boldsymbol{\sigma^2}\mathbf{I_n})$. Our goal is to recover $\boldsymbol{x}_0$ given $\boldsymbol{y}$ and $\boldsymbol{G}_n$. For a given $k$, we consider a maximum *a posteriori* (MAP) estimate of $\boldsymbol{x}_0$ under the signal prior $p_{\boldsymbol{G}_k}$. This results in the following optimization problem over the latent space:

$$\hat{\boldsymbol{z}}_\gamma(k) := \arg\min_{\boldsymbol{z}^k \in \mathbb{R}^k} \frac{1}{2}\left\|\boldsymbol{y} - \boldsymbol{G}_k \begin{pmatrix} \boldsymbol{z}^k \\ 0 \end{pmatrix}\right\|^2 + \frac{\gamma}{2}\|\boldsymbol{z}^k\|^2 \tag{8}$$

where $\begin{pmatrix} \boldsymbol{z}^k \\ 0 \end{pmatrix} \in \mathbb{R}^n$ is the vector $\boldsymbol{z}^k$ padded with zeroes, the estimated signal is $\hat{\boldsymbol{x}}_\gamma(k) = \boldsymbol{G}_k \begin{pmatrix} \hat{\boldsymbol{z}}_\gamma(k) \\ 0 \end{pmatrix}$, and $\boldsymbol{\gamma}$ is a parameter governing how strongly the prior is enforced. The case of $\boldsymbol{\gamma} = 0$ corresponds to a Maximum Likelihood Estimate (MLE) formulation, and the case of $\boldsymbol{\gamma} = \boldsymbol{\sigma}^2$ corresponds to true MAP. It is common in the literature to consider $\boldsymbol{\gamma}$ as a hyperparameter, and thus we study the behavior Equation (8) for all $\boldsymbol{\gamma} \geq 0$.

The following theorem provides an exact expression for the mean square error of the estimate above.

**Theorem 5.1.** *Suppose we have a family $\{\boldsymbol{G}_k\}_{k=1\dots n}$ of generative models as given above, and let $p_{\boldsymbol{G}_k} = \mathcal{N}(0, \boldsymbol{G}_k\boldsymbol{G}_k^T)$, and let $G_n \in \mathbb{R}^{n \times n}$ have singular values $\boldsymbol{s}_1 \geq \boldsymbol{s}_2 \geq \cdots \geq \boldsymbol{s}_n > 0$. Let $\boldsymbol{x}_0 \sim p_{\boldsymbol{G}_n}$ and $\boldsymbol{\eta} \sim \mathcal{N}(0, \boldsymbol{\sigma}^2\boldsymbol{I}_n)$. Then the estimator given by equation 8 yields*

$$\mathbb{E}_{\boldsymbol{x}_0, \boldsymbol{\eta}}\big[\, \|\hat{\boldsymbol{x}}_\gamma(k) - \boldsymbol{x}_0\|^2 \,\big] = \sum_{i=1}^{k} \frac{\boldsymbol{s}_i^2\left(\boldsymbol{s}_i^2\boldsymbol{\sigma}^2 + \boldsymbol{\gamma}^2\right)}{(\boldsymbol{s}_i^2 + \boldsymbol{\gamma})^2} + \sum_{j=k+1}^{n} \boldsymbol{s}_j^2.$$

The exact expression for reconstruction error in Theorem 5.1 makes it possible to immediately read of the optimal value of the tunable complexity parameter $k$, as established in the following corollary.

**Corollary 5.2.** *Under the assumptions of Theorem 5.1, if $\boldsymbol{\gamma} \leq \boldsymbol{\sigma}^2/2$, then the parameter $k$ that leads lowest reconstruction error of the target signal is given by:*

$$\underset{k \in [n]}{\arg\min} \, \mathbb{E}_{\boldsymbol{x}_0, \boldsymbol{\eta}} \|\hat{\boldsymbol{x}}_\gamma(k) - \boldsymbol{x}_0\|^2 = \max\Big\{k \mid \boldsymbol{s}_k \geq \sqrt{\boldsymbol{\sigma}^2 - 2\boldsymbol{\gamma}}\Big\}.$$

This corollary shows that for small enough hyperparameters $\boldsymbol{\gamma}$, and in the presence of sufficient noise, then the optimal complexity parameter for minimizing reconstruction error can be less than the full signal dimensionality. In the case of MLE, an intermediate signal complexity is optimal if the variance of the measurement noise is larger than the smallest singular value of the linear generator $\boldsymbol{G}$. Additionally, we observe theoretically that if the noise level increases, then the optimal value of the tunable complexity decreases. The proofs to Theorem 5.1 and Corollary 5.2 are provided in Appendix.

# 6 Conclusion

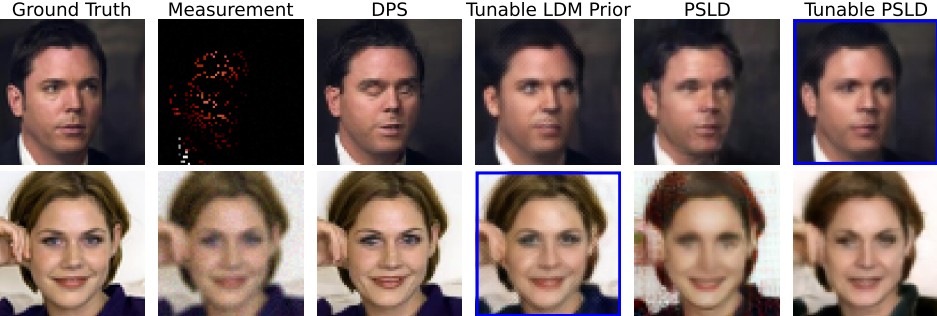

Figure 5: Qualitative results based on Table 2. Blue is lowest LPIPS Score.

Prior work on solving inverse problems with generative priors has primarily focused on fixed models with fixed complexity, with most efforts directed at improving inversion algorithms. In this paper, we introduce tunability of complexity as a complementary and orthogonal axis of improvement. Rather than replacing algorithmic developments, tunability adds a new degree of freedom that practitioners can integrate into their preferred methods with minimal additional training overhead. We show that tunability consistently improves reconstruction across diverse model classes and inverse problems, with intermediate complexity often outperforming both lower and higher extremes. Finally, we present the first nested-dropout training algorithm for latent diffusion models, enabling adaptive complexity in SOTA generative priors. This constitutes a methodological advance for the diffusion community with impact beyond inverse problems.

This work opens several directions for future work. These include extending beyond latent models to pixel-space or other architectures, exploring softer and more flexible forms of complexity control, scaling to very large high-resolution datasets, and developing inversion algorithms specifically tailored to tunable models. Together, these opportunities point to tunability as a general principle for advancing generative priors in inverse problems.

# 7    REPRODUCIBILITY STATEMENT

We report all hyperparameter choices for our method in the Appendix and include detailed descriptions of the setup and configuration for every baseline. All models used in our experiments were trained by us, and to facilitate reproducibility, we will release both the code and trained model checkpoints to the public after the review period.

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

# A  Appendix

We will release our code publicly after the review period. All fid scores are computed with (Parmar et al., 2022).

## A.1  VAE Training/Inversion

Our VAE follows the `AutoencoderKL` architecture from von Platen et al. (2022), with a latent dimensionality of $16 \times 16 \times 16$, resulting in a model of approximately 44 million parameters. The model was trained as described in the main text. For the hyperparameter $\lambda$, we used a fixed value of $0.1$. To incorporate perceptual similarity, we employed a VGG-based perceptual loss. Training was performed with a single $\lambda$ setting and a geometric series success parameter of $10^{-3}$. Further hyperparameter exploration remains possible. Futhermore, for the inversion algorithm refer to (Bora et al., 2017) Github Repository. Simple implementation of gradient descent on the latent vector.

## A.2  LDM Training/Inversion

Following von Platen et al. (2022), we trained a `UNet2DModel` with approximately 200 million parameters using the AdamW optimizer with a learning rate of $1 \times 10^{-4}$ and a batch size of 128. We employed the `DDPMScheduler` provided in the `diffusers` package. The FID scores reported below were generated using the `DPMSolverMultistepScheduler` (Lu et al., 2022) with 100 sampling steps. During training, we withheld a validation set and evaluated the FID score every 20,000 iterations, retaining the three checkpoints with the lowest FID scores. For inverse problems, we used the model corresponding to the best FID score. The FID score was computed using 50,000 training examples, while the test score was evaluated on 10,000 unseen images from the PyTorch implementation of CelebA.

For the inversion algorithms DPS (Chung et al., 2023) and PSLD/LPDS (Rout et al., 2023), we used the official codebases but adapted them to the Diffusers framework, which already provides existing pipelines. All hyperparameters were selected by grid search over the configuration files provided on GitHub, and we report results using the best-performing settings. These search procedures will be released alongside our official code. In general, for tunability we observed that DDIM sampling led to stronger improvements in intermediate representations compared to stochastic samplers, though additional experimental evidence is needed to fully understand this effect. For Algorithm 2, the best-performing setup used the Adam optimizer (Loshchilov & Hutter, 2019) with the DDIM reverse

## FID Score Versus Latent Dimensionality for VAE Reconstructions

Figure 6: Reconstruction FID as function of latent dimensionality k.

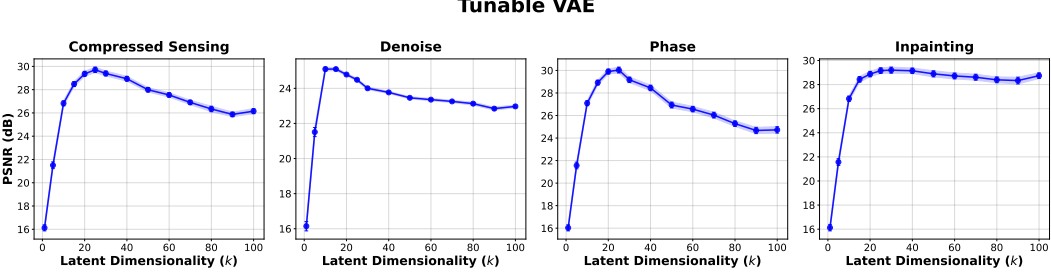

Figure 7: Performance of generative prior with a tunable complexity (Tunable VAE Prior) and its fixed complexity counterpart (VAE) for compressed sensing, inpainting, and denoising on CelebA Dataset. Tunable prior demonstrates a range of tunable parameters $k$ that lead to a better estimate of the target signal measured in PSNR than its baseline.

equation, 5 optimization steps per update, 500 reverse steps, and variance set to 0.1. Importantly, we found that the algorithm does not work reliably without introducing a small degree of stochasticity. This approach is particularly effective in our setting due to the relatively large latent space compared to the image space. The overall objective was to implement a straightforward inversion algorithm that can be readily applied across a wide range of problems.

## B  Normalizing Flow Inverse Problems

We assume the practioner has trained a tunable normalizing flow with the following objective provided by Bekasov & Murray (2020). Our formulation is for latent-variable generative priors. For simplicity assume the measurements have the following form: the forward operator $\mathcal{A} = \boldsymbol{A} \in R^{m \times n}$ and the additive noise term is i.i.d $\eta \sim \mathcal{N}(0, \sigma^2 \boldsymbol{I}_m)$, as in the compressed sensing case. Thus the measurements are generated by $\boldsymbol{y} = \mathbf{A}\boldsymbol{x} + \eta$, where $\boldsymbol{x} \in \mathbb{R}^n$ is an unknown signal.

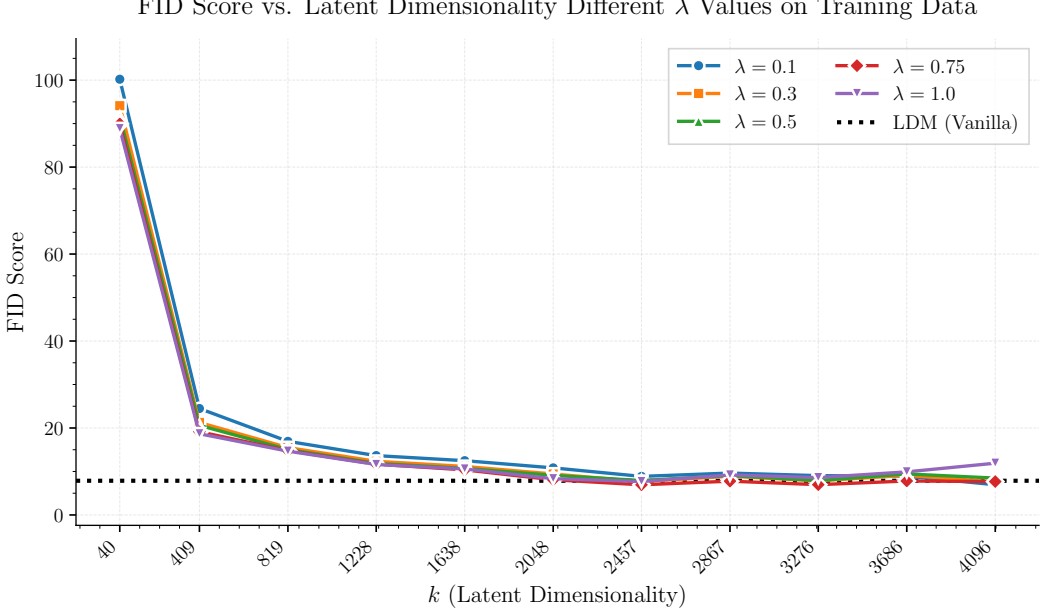

Figure 8: Samples Generated FID Score as a function of k (latent dimensionality)

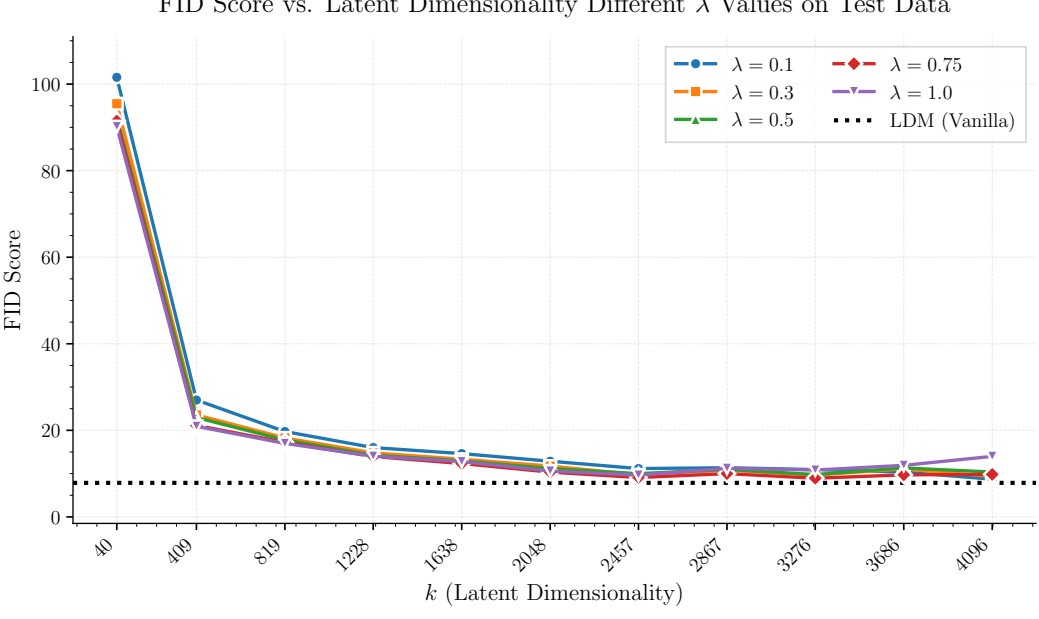

Figure 9: Samples Generated FID Score as a function of k (latent dimensionality)

Consider a fixed invertible neural network $G : \mathbb{R}^n \to \mathbb{R}^n$. Define for each $k \in [n] = \{1, \ldots, n\}$ the map

$$G_k : \mathbb{R}^k \longrightarrow \mathbb{R}^n$$

$$z \longmapsto G\left(\begin{pmatrix} z \\ 0 \end{pmatrix}\right).$$

Method: Gradient Based Sampling  •  Forward op: Phase retrieval (|Ax|) (m/n=0.15)  •  LPIPS (batch avg, best-over-steps): 0.132
Ground Truth                                                    Reconstruction

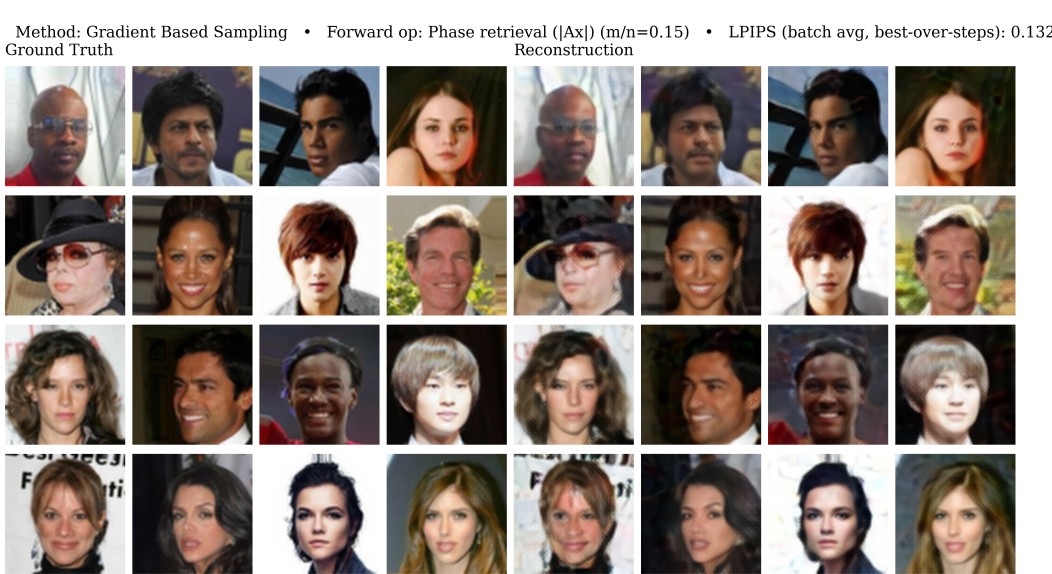

Figure 10: Qualitative Results of Figure 4

Method: Gradient Based Sampling  •  Forward op: Phase retrieval (|Ax|) (m/n=0.15)  •  LPIPS (batch avg, best-over-steps): 0.132
Ground Truth                                                    Reconstruction

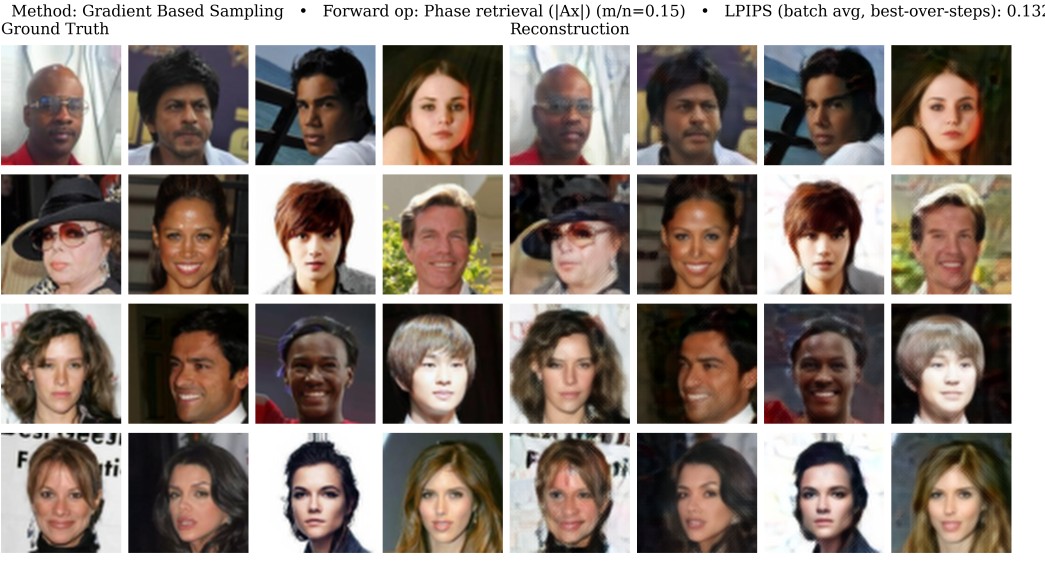

Figure 11: Qualitative Results of Figure 4

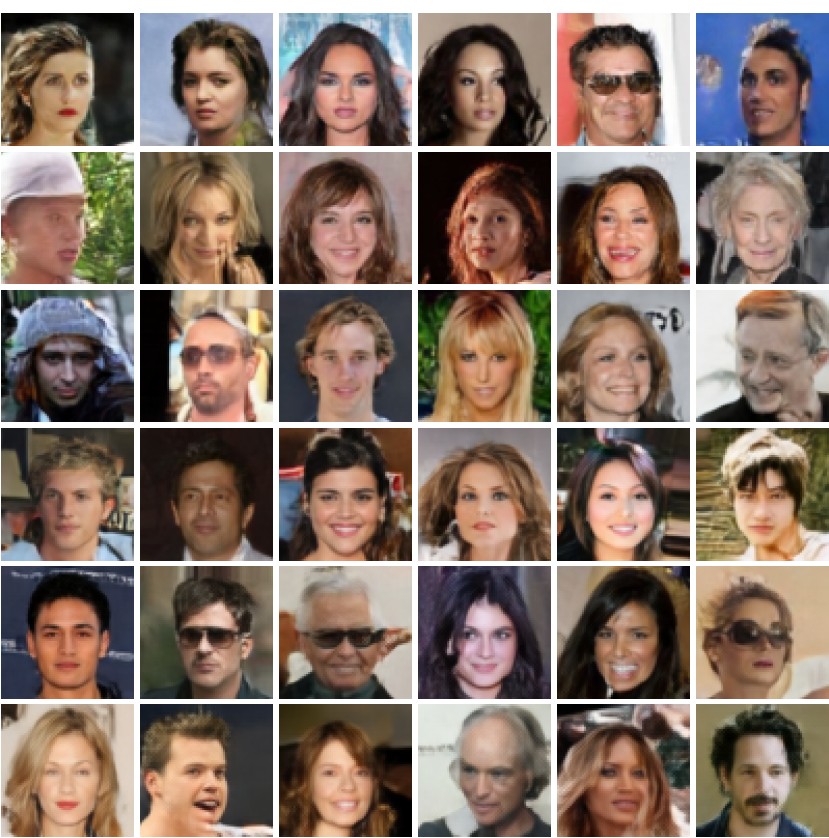

Figure 12: Generated samples from the latent diffusion model with different latent dimensionalities. Each row shows images produced when only the first $k$ of the 4096 latent dimensions are kept during sampling ($k = 4096$: 100%, $k = 3600$: 88%, $k = 3000$: 73%, $k = 2500$: 61%, $k = 2000$: 49%, $k = 1500$: 37%).

We note that $k$ is the parameter that governs the complexity of the prior, and $\begin{pmatrix} \boldsymbol{z} \\ 0 \end{pmatrix} \in \mathbb{R}^n$ is obtained by latent representation $\boldsymbol{z} \in \mathbb{R}^k$. Then the signal representation is given by $x = G((\boldsymbol{z}^T, 0))$ where $x \in \mathbb{R}^n$ is as above.

We are interested in recovering a signal $\boldsymbol{x} \in \mathbb{R}^n$ given a set of noisy measurements $\boldsymbol{y} \in \mathbb{R}^m$. Our prior has a valid density over the entire signal space when $x \in \text{Range}(G_k)$, therefore a natural attempt to solve the given inverse problems is by a maximum *a posteriori* estimation:

$$
\begin{aligned}
\hat{x}_{\text{MAP}}(k) &:= \underset{\boldsymbol{x} \in \text{Range}(G_k)}{\arg\min} \; -\log p(\boldsymbol{x}|\boldsymbol{y}) \\
&= \underset{\boldsymbol{x} \in \text{Range}(G_k)}{\arg\min} \; -\log p(\boldsymbol{y}|\boldsymbol{x}) - \log p_{G_k}(x) \\
&= \underset{x \in \text{Range}(G_k)}{\arg\min} \; \frac{1}{2}\|\boldsymbol{y} - \boldsymbol{A}\boldsymbol{x}\|^2 - \sigma^2 \log p_{G_k}(x)
\end{aligned}
$$

where $p_{G_k}$ is the density function on $x$ induced by $G_k$, $\|\cdot\|$ is the Euclidean norm, and $\sigma$ is given by the model noise. Similar to previous work of Asim et al. (2020); Whang et al. (2021), we optimized over the latent space given by:

$$
\hat{\boldsymbol{z}}_{\text{MAP}}(k) := \underset{\boldsymbol{z} \in \mathbb{R}^k}{\arg\min} \; \frac{1}{2} \left\| \boldsymbol{y} - \boldsymbol{A}G\left(\begin{pmatrix} \boldsymbol{z} \\ 0 \end{pmatrix}\right) \right\|^2 - \sigma^2 \log p_{G_k}\left(G\left(\begin{pmatrix} \boldsymbol{z} \\ 0 \end{pmatrix}\right)\right)
$$

The proposed optimization problem above is solved via gradient descent with Adam optimizer Kingma & Ba (2015) and is initialized at $z = 0$. Depending on the specific inverse problem at hand the tunable parameter $k$ can be selected to change the dimensionality of the model, so the first $k$ elements of the vector $z$ will be optimized over and the rest will be set to zero. Our formulation relies on a density function $p_{G_k}$, and works by Asim et al. (2020); Whang et al. (2021) have shown that a smoothing parameter on a density function improves performance when solving the optimization problem above. We empirically observe the same phenomena, and as a result we replace $\sigma^2$ with a hyperparameter $\gamma$. For more information about choosing hyperparameters and how to compute $p_{G_k}$ please refer to the Appendix. Therefore instead we optimized over a modified MAP estimate given by

$$
\hat{z}_{\text{MAP}}(k) := \underset{z \in \mathbb{R}^k}{\arg\min} \; \frac{1}{2} \left\| y - AG\left(\begin{pmatrix} \boldsymbol{z} \\ 0 \end{pmatrix}\right) \right\|^2 - \gamma \log p_{G_k}\left(G\left(\begin{pmatrix} \boldsymbol{z} \\ 0 \end{pmatrix}\right)\right).
$$

For the VAE we follow the same formulation as Bora et al. (2017), which is similar to the one above, but instead of having a tractable density we approixmate by with $\ell_2$ penality in the latent space.

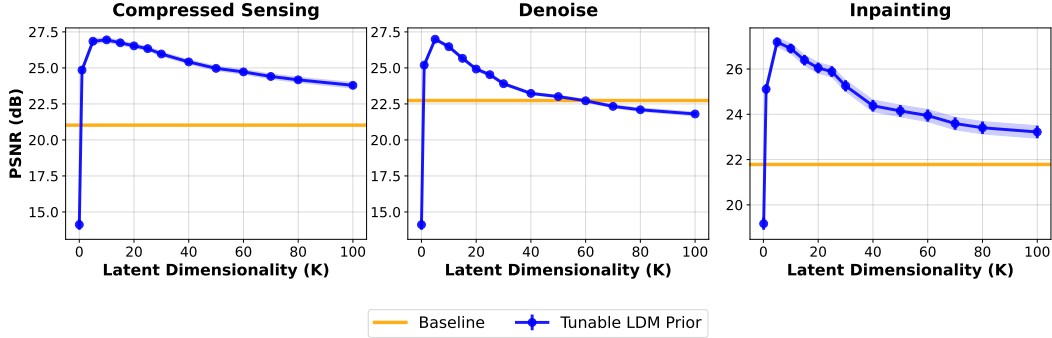

**CelebA NF 5-bit Results**

Figure 14: Performance of generative prior with a tunable complexity (Tunable NF Prior) and its fixed complexity counterpart (Normalizing Flow) for compressed sensing, inpainting, deblurring, and denoising on CelebA Dataset. Tunable prior demonstrates a range of tunable parameters $k$ that lead to a better estimate of the target signal measured in PSNR than its baseline.

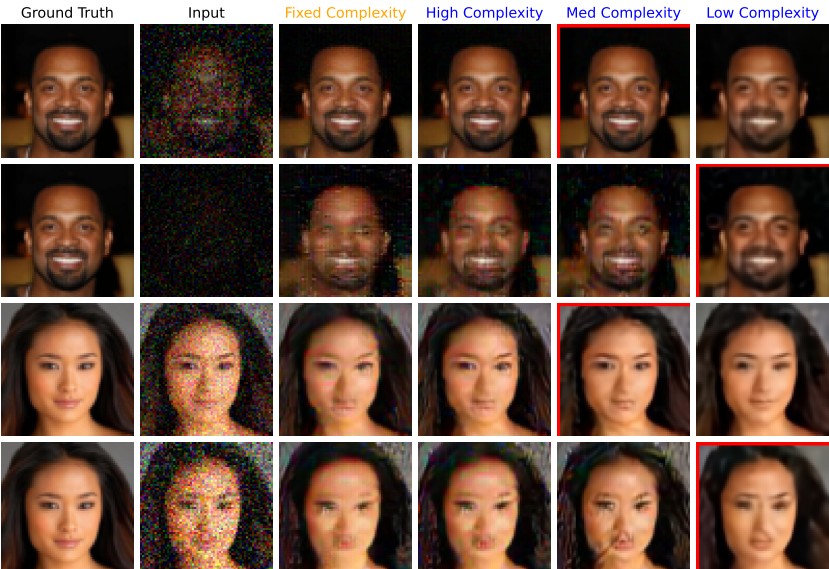

Figure 13: Results on inpainting and denoising across various measurement regimes for Normalzing Flow. The tuning parameter should be chosen based on the complexities of the given inverse problem. The red bounding box indicates which model achieved the lowest reconstruction error with respect to SSIM.

# C   Proofs

*Proof of Theorem 5.1.*

Without loss of generality, we take $G_k$ to be a diagonal matrix. This follows because of invariance of Euclidean norms with respect to orthorgonal transformations and because of the rotational invariance of the random variable $z_0$. Let $\mathcal{P}_k$ and $\mathcal{P}_k^\perp$ be the orthogonal projectors onto $\mathrm{span}(\{e_1, \ldots, e_k\})$ and its orthogonal complement, respectively. For a diagonal matrix $\Sigma = \mathrm{diag}(s_1, s_2, \cdots, s_n) \in \mathbb{R}^{n \times n}$, let $\boldsymbol{S}_k \in \mathbb{R}^{k \times k}$ and $\boldsymbol{S}_k^\perp \in \mathbb{R}^{n-k \times n-k}$ be such that

$$\Sigma = \begin{pmatrix} \boldsymbol{S}_k & \boldsymbol{0} \\ \boldsymbol{0} & \boldsymbol{S}_k^\perp \end{pmatrix}.$$

We want to find the expectation

$$\mathbb{E}_{x_0, \eta} \|\hat{\boldsymbol{x}}_\gamma(k) - x_0\|^2 = \mathbb{E}_{x_0, \eta} \|\mathcal{P}_k(\hat{\boldsymbol{x}}_\gamma(k) - x_0)\|^2 + \mathbb{E}_{x_0} \|\mathcal{P}_k^\perp(x_0)\|^2. \tag{9}$$

For the equation above we will compute the first term on the right hand side first, then do the same for the second term on the right hand side. The first term on the right hand side breaks into sum of two terms which can readily computed, while the second term on the right hand side follows from a lemma. Notice $\mathcal{P}_k(\hat{\boldsymbol{x}}_\gamma(k)) = \boldsymbol{S}_k \hat{\boldsymbol{z}}_\gamma(k) \in \mathbb{R}^k$ and the solution of the optimization problem equation 8 is

$$\hat{\boldsymbol{z}}_\gamma(k) := (\boldsymbol{S}_k \boldsymbol{S}_k^T + \gamma \boldsymbol{I}_k)^{-1} \boldsymbol{S}_k^T \mathcal{P}_k(y)$$
$$= (\boldsymbol{S}_k \boldsymbol{S}_k^T + \gamma \boldsymbol{I}_k)^{-1} \boldsymbol{S}_k^T (\boldsymbol{S}_k \mathcal{P}_k(z_0) + \mathcal{P}_k(\eta)).$$

Thus,

$$
\begin{aligned}
\mathbb{E}_{x_0,\eta}\|\mathcal{P}_k(\hat{\boldsymbol{x}}_\gamma(k) - x_0)\|^2 &= \mathbb{E}_{z_0,\eta}\|\boldsymbol{S}_k\hat{\boldsymbol{z}}_\gamma(k) - \boldsymbol{S}_k\mathcal{P}_k(z_0)\|^2 \\
&= \mathbb{E}_{z_0,\eta}\|\boldsymbol{S}_k(\boldsymbol{S}_k\boldsymbol{S}_k^T + \gamma\boldsymbol{I}_k)^{-1}\boldsymbol{S}_k^T\mathcal{P}_k(y) - \boldsymbol{S}_k\mathcal{P}_k(z_0)\|^2 \\
&= \mathbb{E}_{z_0,\eta}\|\boldsymbol{S}_k(\boldsymbol{S}_k\boldsymbol{S}_k^T + \gamma\boldsymbol{I}_k)^{-1}\boldsymbol{S}_k^T(\boldsymbol{S}_k\mathcal{P}_k(z_0) + \mathcal{P}_k(\eta)) \\
&\quad - \boldsymbol{S}_k\mathcal{P}_k(z_0)\|^2 \\
&= \mathbb{E}_{z_0,\eta}\|(\boldsymbol{S}_k(\boldsymbol{S}_k\boldsymbol{S}_k^T + \gamma\boldsymbol{I}_k)^{-1}\boldsymbol{S}_k - \boldsymbol{S}_k)\mathcal{P}_k(z_0) \\
&\quad + \boldsymbol{S}_k(\boldsymbol{S}_k\boldsymbol{S}_k^T + \gamma\boldsymbol{I}_k)^{-1}\boldsymbol{S}_k^T\mathcal{P}_k(\eta)\|^2
\end{aligned}
\tag{10}
$$

The last term above Equation (10) can be decomposed

$$
\mathbb{E}_{z_0}\left\|\left(\boldsymbol{S}_k(\boldsymbol{S}_k\boldsymbol{S}_k^T + \gamma\boldsymbol{I}_k)^{-1}\boldsymbol{S}_k - \boldsymbol{S}_k\right)\mathcal{P}_k(z_0)\right\|^2
\tag{11}
$$

$$
+ \mathbb{E}_{z_0,\eta}\left\langle\left(\boldsymbol{S}_k(\boldsymbol{S}_k\boldsymbol{S}_k^T + \gamma\boldsymbol{I}_k)^{-1}\boldsymbol{S}_k - \boldsymbol{S}_k\right)\mathcal{P}_k(z_0), \boldsymbol{S}_k(\boldsymbol{S}_k\boldsymbol{S}_k^T + \gamma\boldsymbol{I}_k)^{-1}\boldsymbol{S}_k^T\mathcal{P}_k(\eta)\right\rangle
\tag{12}
$$

$$
+ \mathbb{E}_\eta\left\|\boldsymbol{S}_k(\boldsymbol{S}_k\boldsymbol{S}_k^T + \gamma\boldsymbol{I}_k)^{-1}\boldsymbol{S}_k^T\mathcal{P}_k(\eta)\right\|^2
\tag{13}
$$

For the term above, the expectation of inner product term (Equation (12)) is zero becasue both random variables have an expectation of zero. Futhermore, apply Lemma C.1 to Equation (11), yielding

$$
\begin{aligned}
\mathbb{E}_{z_0}\|(\boldsymbol{S}_k(\boldsymbol{S}_k\boldsymbol{S}_k^T + \gamma\boldsymbol{I}_k)^{-1}\boldsymbol{S}_k^T\boldsymbol{S}_k - \boldsymbol{S}_k)\mathcal{P}_k(z_0)\|^2 \\
= \|(\boldsymbol{S}_k\boldsymbol{S}_k\boldsymbol{S}_k^T + \gamma\boldsymbol{I}_k)^{-1}\boldsymbol{S}_k^T\boldsymbol{S}_k - \boldsymbol{S}_k\|_F^2 \\
= \sum_{i=1}^k\left|\frac{s_i^3}{s_i^2 + \gamma} - s_i\right|^2 \\
= \sum_{i=1}^k\frac{s_i^2\gamma^2}{(s_i^2 + \gamma)^2}.
\end{aligned}
\tag{14}
$$

Then apply Lemma C.2 to Equation (13)

$$
\begin{aligned}
\mathbb{E}_\eta\|\boldsymbol{S}_k(\boldsymbol{S}_k\boldsymbol{S}_k^T + \gamma\boldsymbol{I}_k)^{-1}\boldsymbol{S}_k^T\mathcal{P}_k(\eta)\|^2 \\
= \text{Tr}(\boldsymbol{S}_k(\boldsymbol{S}_k\boldsymbol{S}_k^T + \gamma\boldsymbol{I}_k)^{-1}\boldsymbol{S}_k^T\sigma^2\boldsymbol{I}_k\boldsymbol{S}_k(\boldsymbol{S}_k\boldsymbol{S}_k^T + \gamma\boldsymbol{I}_k)^{-1}\boldsymbol{S}_k^T) \\
= \sum_{i=1}^k\frac{s_i^4\sigma^2}{(s_i^2 + \gamma)^2}.
\end{aligned}
\tag{15}
$$

This concludes the calculation of the first term on the right hand side of Equation (9). Now we apply Lemma C.1 to the second term on the right hand side

$$
\mathbb{E}_{x_0}\|\mathcal{P}_k^\perp(x_0)\|^2 = \mathbb{E}_{z_0}\|\boldsymbol{S}_k^\perp\mathcal{P}_k^\perp(z_0)\|^2 = \|\boldsymbol{S}_k^\perp\|_F^2 = \sum_{j=k+1}^n s_j^2.
\tag{16}
$$

Lastly, combine Eq.16,14,15 and simplify

$$
\mathbb{E}_{x_0,\eta}\|\hat{\boldsymbol{x}}_\gamma(k) - x_0\|^2 = \sum_{i=1}^k\frac{s_i^2(s_i^2\sigma^2 + \gamma^2)}{(s_i^2 + \gamma)^2} + \sum_{j=k+1}^n s_j^2.
$$

$\square$

*Proof of Corollary 5.2.*
For notational convenience, let $E(k) = \mathbb{E}_{x_0,\eta}\|\hat{x}_\gamma(k) - x_0\|^2$. Since $[n]$ is a finite non-empty set, $\arg\min_k E(k)$ exists, and so we consider the set $C$ of *candidate* minimizers,

$$
C = \{k \mid E(k-1) \geq E(k)\}.
$$

Notice $E(k-1) - E(k) \geq 0$ for each $k \in C$, and moreover that this difference is

$$\sum_{i=1}^{k-1} \frac{s_i^2(s_i^2\sigma^2+\gamma^2)}{(s_i^2+\gamma)^2} + \sum_{j=k}^{n} s_j^2 - \sum_{i=1}^{k} \frac{s_i^2(s_i^2\sigma^2+\gamma^2)}{(s_i^2+\gamma)^2}$$

$$- \sum_{j=k+1}^{n} s_j^2 \geq 0$$

$$\iff$$

$$s_k - \frac{s_k^2(s_k^2\sigma^2+\gamma^2)}{(s_k^2+\gamma)^2} \geq 0$$

$$\iff$$

$$s_k \geq \sqrt{\sigma^2 - 2\gamma},$$

which reveals

$$C = \{k \mid s_k \geq \sqrt{\sigma^2 - 2\gamma}\}.$$

From our assumptions, $\sigma^2 - 2\gamma \geq 0$. Note now that for any $i, j \in C$ with $i < j$, we have that

$$s_j \geq s_k \geq s_i \geq \sqrt{\sigma^2 - 2\gamma} \quad \text{for all } i < k \leq j,$$

hence $k \in C$.

To prove the corollary is to prove that $\arg\min_k E(k) = \max C$, so it suffices to show $E$ is decreasing on $C$. Since $s_i^2 \geq \sigma^2 - 2\gamma$ for all $i \in C$, it is clear that

$$s_i^4 + s_i^2(2\gamma - \sigma^2) \geq s_i^4 - s_i^2 \cdot s_i^2 = 0$$

is non-negative. From this, we deduce that $s_i^2\sigma^2 + \gamma^2 \leq (s_i^2+\gamma)^2$, hence

$$\Delta(i) := 1 - \frac{s_i^2\sigma^2+\gamma^2}{(s_i^2+\gamma)^2} \geq 0$$

is also non-negative.

Let $P, Q \in C$ be such that $P \leq Q$. We'll show $E(P) \geq E(Q)$ by showing the difference $E(P) - E(Q)$ is non-negative. It is easily computed by

$$E(P) - E(Q) = \sum_{i=P+1}^{Q} s_i^2 \Delta(i).$$

Previous work shows that $\Delta(i)$ is non-negative for all $i$ in the range $P < i \leq Q$ since all such $i$ are elements of $C$, thus concluding the proof.

$\square$

**Lemma C.1.** *Let $x \sim \mathcal{N}(0, \boldsymbol{I}_n) \in \mathbb{R}^n$ then for any matrix $M \in \mathbb{R}^{m \times n}$, we have*

$$\mathbb{E}_x \|Mx\|^2 = \|M\|_F^2.$$

*Proof of Lemma C.1.*
Compute that

$$\begin{aligned}
\mathbb{E}_x \|Mx\|^2 &= \mathbb{E}_x \langle Mx, Mx \rangle \\
&= \mathbb{E}_x \langle M^T M, xx^T \rangle \\
&= \langle M^T M, \boldsymbol{I}_n \rangle \\
&= \|M\|_F^2.
\end{aligned}$$

$\square$

**Lemma C.2.** *Let $x \sim \mathcal{N}(0, \Sigma) \in \mathbb{R}^n$ then for any matrix $M \in \mathbb{R}^{m \times n}$, we have*

$$\mathbb{E}_x \|Mx\|^2 = Tr(M\Sigma M^T).$$

*Proof of Lemma C.2.*
Let $Y = Mx$, then $Y \sim \mathcal{N}(0, M\Sigma M^T)$. Thus,

$$\mathbb{E}_x \|Mx\|^2 = \mathbb{E}_Y \|Y\|^2 = \mathbb{E}[Y^T Y] = \sum_i^n Y_i^2 = \sum_{i=1}^n \text{Var}(Y_i)$$

$$= \text{Tr}(M\Sigma M^T). \tag{17}$$

$\square$

You may include other additional sections here.