# OpenReview forum: "Latent Generative Models with Tunable Complexity for Compressed Sensing and other Inverse Problems"
_ICLR.cc/2026/Conference — Submitted to ICLR 2026_

### Official Review · Reviewer_b6w2 · 2025-10-21

**Soundness:** 2
**Presentation:** 3
**Contribution:** 3
**Rating:** 4
**Confidence:** 4

**Summary:**

The paper proposes tunable-complexity generative priors for inverse problems such as compressed sensing, inpainting, denoising, and phase retrieval. The authors claim that traditional generative models (GANs, VAEs, diffusion models, normalizing flows) have fixed latent dimensionality, which limits adaptability. To address this, they propose a framework where model complexity (latent dimension k) can be tuned at inference time, without retraining. They demonstrate this idea in three classes of generative models — VAE, normalizing flow, and latent diffusion model (LDM) — using nested dropout to enforce hierarchical latent representations. Empirical improvements on multiple inverse problems. A simple theoretical justification  demonstrating why an intermediate latent dimensionality minimizes MSE in linear denoising.

**Strengths:**

1. The paper is well-written and well-organized.
2. The contribution is novel and interesting. The idea of tunable generative prior for solving inverse problems is novel.
3. The proposed method is unified and applied for three classes of generative priors (diffusion models, normalizing flows, and VAE).

**Weaknesses:**

1. While the theoretical results provided are valuable, the scope is narrow and assumptions are strong. It only is applied to Gaussian denoising with a linear invertible generative model, which is limiting and does not provide additional insight to a general class of inverse problems.

2. The provided experimental results are limited and does not provide enough evidence to support the claims.
* The method seems to be not working in the case of normalizing flow in case of inpainting and Gaussian blur.
* All experiments are on CelebA 64×64; no larger or more diverse datasets (e.g., ImageNet, FFHQ) are tested.
* The nested-dropout training’s computational overhead, convergence behavior, and sensitivity to λ parameters are not analyzed. It is not clear what is the additional cost of the proposed training.
* There is not enough experimental results to verify the claims throughly. For example, each of the generative models need to have a visual and a table comparing the proposed prior with SOTA methods. The visuals must include the quantitative metric such as LPIPS, PSNR, and FID. There is only two tables for NF, which one is showing suboptimal performance.
* Also, the details of the prior used and the baseline should be included in the caption of the figures.

**Questions:**

Including a more through experimental analysis would strengthen the papers greatly.

---

### Official Review · Reviewer_PwFT · 2025-10-31

**Soundness:** 2
**Presentation:** 2
**Contribution:** 1
**Rating:** 2
**Confidence:** 4

**Summary:**

This paper introduces tunable-complexity generative priors for ill-posed inverse problems. The authors demonstrate their approach across three families of generative models, normalizing flows, variational autoencoders (VAEs), and diffusion models. Experiments on simulated data and forward models indicate that the proposed method yields modest but consistent improvements in reconstruction quality across several inverse problems.

**Strengths:**

The paper is well-written and easy to understand.

**Weaknesses:**

The paper has several limitations that make it unsuitable for acceptance at this conference:

1. If I understand correctly, Algorithms 1 and 2 require the tunable parameter  as input, which is not known a priori. Identifying an appropriate value needs running the algorithm many times across different values for each new measurement which is expensive and slow. Beyond the computational cost, this approach is impractical for real-world scientific problems, where the ground truth is unknown and no reliable metric may exist to evaluate reconstruction quality. Please clarify this drawback and clearly discuss these limitations.


2. All experiments are conducted on simulated datasets and forward operators, which do not show the full challenges of real-world inverse problems. Without experiments on real data, it is difficult to assess the practical performance of the proposed method.


3. The use of low-dimensional representations as priors for ill-posed inverse problems has been well established in prior work (e.g., [1]). The authors should provide a more thorough discussion of these related studies to properly discuss the main contribution of this paper.


[1] Kothari, Konik, et al. Trumpets: Injective flows for inference and inverse problems. Uncertainty in Artificial Intelligence. PMLR, 2021.

**Questions:**

Please see the weaknesses.

---

### Official Review · Reviewer_wEw5 · 2025-11-06

**Soundness:** 3
**Presentation:** 3
**Contribution:** 3
**Rating:** 6
**Confidence:** 5

**Summary:**

This paper proposes tunable-complexity generative priors for inverse problems, so the latent dimensionality $k$ can be selected at inference time instead of being fixed at training. The authors instantiate this for VAEs, normalizing flows, and especially latent diffusion models (LDMs) by training with nested dropout to enforce an ordered latent space; they also give a simple posterior-sampling template (Algorithms 1–2) that truncates latents to the first $k$ coordinates during every reverse step and data-consistency update. Empirically, across compressed sensing, inpainting, denoising, and phase retrieval (CelebA 64×64), intermediate $k$ consistently outperforms the fixed baselines, showing an “upside-down U” dependence on $k$. Theoretically, for a linear generator $G$ and Gaussian noise, the paper derives a closed-form MSE for MLE/MAP estimates and shows that the optimal $k$ decreases with noise, giving a concrete rule in terms of singular values of $G$.

**Strengths:**

- Clear problem & useful knob. The paper cleanly articulates the mismatch of fixed latent capacity and inverse-problem difficulty and provides a practical knob $k$ to trade bias/variance at test time.
- Method is simple & broadly applicable. Nested-dropout training + a per-step truncation works across VAEs/NFs/LDMs and multiple inversion solvers (LDPS-style, DPS-style), making adoption easy.
- Consistent empirical gains. Plots and tables show nontrivial improvements over fixed-$k$ baselines on CS, inpainting, deblurring/denoising, and phase retrieval; NF variants show large boosts.
- Theory that matches intuition. The MSE formula and corollary explicitly tie optimal $k$ to noise and singular spectrum, rationalizing the “medium-$k$” phenomenon.
- Positioning vs. literature. The work complements established inversion with diffusion (DPS, LDPS/PSLD, ReSample) by adding a new axis (capacity tuning) rather than another solver.

**Weaknesses:**

- Scope & datasets. Experiments are mainly CelebA 64×64; it’s unclear if tunability holds at higher resolutions or domains (MRI/CT, natural high-res). External SOTA pixel-space diffusion priors (DPS variants) are compared but breadth is limited.

- Existing theorem in literature. Theorem 1 in Asim et al (https://arxiv.org/pdf/1905.11672) has a very similar result in terms of the singular values of $G$. It's also closely related to the main theorems in Yu and Shapiro (https://arxiv.org/pdf/1101.5785)

- Metric choices & consistency. Some plots use LPIPS while others use PSNR; selection sometimes favors the proposed method (e.g., LPIPS better for LDMs, PSNR for NF). A unified metric suite and confidence intervals across all tasks would strengthen claims.

- Ablations on training hyper-params. The nested-dropout distribution $p_k$ and the interpolation weight $\lambda$ (Eq. 7) are pivotal; while the paper shows FID vs $k$ for several $\lambda$, it lacks ablations on $p_k, \lambda$ schedules, and how these choices shift the best $k$ for downstream tasks.

- Theory–practice gap. The theory is for linear generators; guidance to map that to modern LDMs (nonlinear decoders, score fields) is qualitative. A partial bridge (e.g., linear with ReLUs) would help.


- Positioning vs prior solvers. While the paper cites DPS/PSLD/ReSample, head-to-head comparisons vary by operator; clearer, apples-to-apples settings mirroring those papers would aid attribution of gains to tunability alone.

**Questions:**

- I'm happy to increase my score if you can show experiments on a harder problem (such as MRI with diffusion) or higher resolution (FFHQ 1024x1024). Compressed sensing with CelebA is too small a dataset to make solid conclusions.

- Automatic $k$ selection. Can you propose a data-driven rule (e.g., early-stop validation loss in latent posterior steps, SURE, or discrepancy principle) to pick $k$ per instance, not per task? How does it compare to the corollary’s singular-value rule in practice?

- Generalization across operators. Do the optimal $k$s transfer between CS and inpainting for the same image class, or is per-operator tuning required? Any signs of overfitting to the operator during solver hyper-tuning?

- Nested-dropout design. How sensitive are results to the truncated geometric $p_k$? Did you try deterministic curricula (monotone $k$ scheduling) or temperature-annealed $\lambda$ in Eq. (7)?

- Robustness. How stable is Algorithm 2 to misspecified noise levels and forward-model mismatch? Any failure cases where small $k$ collapses texture or large $k$ overfits noise?

---

### Official Review · Reviewer_xYbV · 2025-11-11

**Soundness:** 3
**Presentation:** 3
**Contribution:** 2
**Rating:** 4
**Confidence:** 4

**Summary:**

The goal of this work is to design a generative neural network prior that can be used to solve linear imaging inverse problems such as compressed sensing, denoising, and phase retrieval. The authors propose a method to train a latent space diffusion model with a hierarchical, sequentially organized structure enforced on the coordinates of the latent code. They also introduce a penalization method for training the proposed latent diffusion algorithm in an "end-to-end" fashion, jointly training an encoder, decoder, and latent diffusion model. The benefit of this approach is that the resulting latent diffusion sampler can be used for downstream inverse problems by exploiting a well-known connection to MAP estimation, which is standard in the literature on solving imaging inverse problems with downstream priors.

To enforce the desired latent code structure, the authors perform training with "nested dropout," in which suffixes of random length in the latent code are masked with some probability during training.

The authors demonstrate through a variety of compressed sensing experiments that the proposed method can be used as a "drop in" improvement to some important existing works in the literature, notably DPS/Latent DPS and Normalizing Flow priors. This is intuitive, because the proposed method subsumes previous methods and so should directly improve on them.

Finally, the authors study the effect of latent code masking in a stylized recovery problem with a linear generator and data generation model. The theory suggests that indeed, partial latent code masking can improve recovery MSE by acting as a regularizer, playing a role complementary to that of ridge regularization.

**Strengths:**

1. The method is well explained and conceptually easy to understand.
2. While I have some reservations about the experiments in this work, they are generally well-designed. I am supportive of the baselines presented in this work, which are outperformed by the proposed method with a nontrivial margin.
3. Theorem 5.1 gives a presents a very simple and intuitive picture for why this method works in the context of regularized linear regression. In the linear setting, Corollary 5.2 directly identifies a threshold index $k$ beyond which the bottom $(k+1, \ldots, n)$ principle components of the data are washed out by noise, such that including these components leads to overfitting. The estimator works by controlling model complexity to mitigate overfitting.

**Weaknesses:**

1. While the authors have selected reasonable baselines to compare to, I am worried that they only present direct comparison results to other baselines at specific measurement indices. For instance, Table 1 uses $m/n=0.15$ and Table 2 uses $m/n = 0.075$. It is possible that the proposed method outperforms competitors at specific measurement indices, but not across the spectrum. This is my primary concern with the work.
2. Further, the results presented in Table 1 and Table 2 appear to be computed using the best performing latent code dimension. In reality, this parameter should be selected using the training data through a process like cross validation. If the authors are indeed reporting only the best validation performance after optimizing over $k$, it becomes an unfair experiment. (e.g. why not then compare to MAP estimation with optimal ridge regularization?).
3. The theoretical results are relatively simple and essentially amount to a calculation that takes place on page 20. While these arguments support a very intuitive picture for why the proposed method is successful, they are not really necessary to develop that intuition and don't contribute much to the overall work. One way this example could provide interesting insights is by directly comparing the regularizing effects of latent masking vs. ridge regularization, since according to Corollary 5.1, if you optimally select the ridge parameter ($\gamma = \sigma^2/2$) then there should be no masking at all. Can latent code masking be seen as a crude alternative to well-tuned ridge regularization?

**Questions:**

1. In Figure 4, it appears that the optimal latent dimensionality occurs at around $k=500$ in all of these inverse problems. This calls into question the value of the proposed model. If there is some kind of "intrinsic dimensionality" effect, such that modeling the true data distribution requires only a low-dimensional latent code, then it would be much cheaper computationally to train a lower-dimensional model and to not bother with inference time tunability for latent code dimension. Are there any recovery settings in your experiments where the optimal $k$ differs by a significant amount from what is shown in Figure 4? What is the performance of signal recovery with just a fixed $k=600$ dimensional latent code, does the proposed method significantly outperform that?
2. How did you select $k$ for the experiments in Table 1, 2?

---

### Author Response · Authors · 2025-12-03
**General Remark**

We thank the reviewers for their thoughtful comments and constructive feedback. We would also like to thank the Area Chair for ensuring a smooth and fair evaluation process despite the unusual circumstances this year.

---

### Author Response · Authors · 2025-12-03
**Experimental Critiques**

Several reviewers raised a common concern regarding the scope of the main experiments, which primarily use CelebA and CelebA-HQ at 64x64 resolution. As an academic lab, our computational budget at the time limited the scale of experiments we could run across each latent variable model considered (VAE, Normalizing Flows, and Latent Diffusion Models). In that setting, 64x64 provided an appropriate comparison point that allowed us to benchmark all model classes fairly and consistently.

To further address this concern, we additionally trained higher-resolution models and evaluated them on FFHQ at 256x256. The results are provided in the following document:
[https://files.catbox.moe/9rpgs3.pdf](https://files.catbox.moe/9rpgs3.pdf)

In these supplementary experiments, we trained the autoencoder for approximately three GPU-days on an A100 (and five GPU-days using mixed precision). The model achieves an FID of roughly 10 on a held-out validation set of 5,000 images. We evaluated three inverse problems using the same Algorithm 2 as shown in Figure 4 of the paper:

* **Denoising:** Gaussian noise with sigma = 0.3.
* **Compressed sensing:** Circulant measurement matrices with a 5% measurement ratio.
* **Phase retrieval:** Undersampled Fourier measurements with a 5% sampling ratio.

We intentionally included extreme regimes in the paper to illustrate the strength of tunability: these regimes make clear where fixed-complexity methods break, and highlight how another member of the tunable family performs substantially better under the same conditions. This is particularly notable given that PSLD and DPS are contemporary state-of-the-art methods; as shown in Tables 1 and 2, the tunable prior sometimes outperforms the fixed-complexity baseline precisely in settings where the latter degrades.

---

### Author Response · Authors · 2025-12-03
**How is tunable parameter k selected?**

In this paradigm, the generative model is trained independently of the forward operator, and the training objective differs from the inversion objective. During training, the goal is for the model to preserve its ability to represent the natural signal class across a broad range of latent dimensionalities, as illustrated in Figure 3, where FID remains essentially constant over a wide span of latent dimensions.

During inversion, however, the latent dimensionality that yields the best reconstruction can vary depending on the forward operator and even on the specific measurement regime associated with that operator. This effect is demonstrated in the supplementary figures:

* Compressed sensing with Gaussian measurements on the CelebA 64x64 test set:
[https://files.catbox.moe/x8gx04.pdf](https://files.catbox.moe/x8gx04.pdf)

* Denoising with Gaussian noise:
[https://files.catbox.moe/10b48a.pdf](https://files.catbox.moe/10b48a.pdf)

Consequently, we treat the choice of latent dimensionality k as a hyperparameter selected via validation for each inverse problem and operator. One of the key advantages of tunability is precisely this ability to adapt the model’s effective complexity to the specific inverse problem at hand.

---

### Author Response · Authors · 2025-12-03
**Concerns about how training hyperparameter can affect results downstream when solving inverse problems**

Regarding concerns about Figure 3 and the choice of the nested dropout probability pk and loss scaling parameter lambda: We acknowledge that these are empirical hyperparameter choices. However, our main experiments demonstrate that the settings used in the paper are sufficient to achieve performance competitive with contemporary state-of-the-art methods, indicating the method is effective even without extensive tuning. We will add an ablation study regarding these parameters to the camera-ready version.

---

### Author Response · Authors · 2025-12-03
**Theory Critique**

The purpose of the theory section is to show the effect in the simplest possible linear setting. When the solution is restricted to lie in a k-dimensional linear manifold, the reconstruction error is not monotone in k; instead, there is an intermediate value of k that achieves the lowest error. This matches the empirical behavior we observe when tuning the latent dimensionality.

This behavior does not appear in ridge regression. Ridge always operates in the full ambient dimension and only shrinks coefficients; it changes the estimator’s bias but not its effective model capacity. Because of this, ridge cannot exhibit a “best intermediate complexity’’ point.

Our calculation also differs from prior theoretical work. Asim et al. provide lower bounds, and Yu and Sapiro compute expectations, but neither examines how reconstruction error changes across a family of models with varying dimensionality. In contrast, our closed-form expression makes it possible to explicitly identify the value of k that minimizes the error in this linear example.

---

### Meta-Review · Area_Chair_S5a6 · 2026-01-06

**Summary:**

The manuscript considers generative prior for inverse problems with a tunable complexity parameter, such as the latent dimension. The paper is largely empirical and requires polishing (e.g., the abstract on openreview is incomplete, the manuscript itself contains contents from the template that is not removed). The authors response also does not fully addressed the concerns raised. Therefore the metareviewer cannot recommend acceptance.

**Reviewer Concerns:**

The authors provide further numerical experiments, while the theory is still lacking.

**Reviewer Scores:**

I believe the reviewers will maintain their scores.

---

### Decision · Program_Chairs · 2026-01-26

Reject